# Impact of Mobile Health Literacy, Stroke-Related Health Knowledge, Health Beliefs, and Self-Efficacy on the Self-Care Behavior of Patients with Stroke

**DOI:** 10.3390/healthcare12191913

**Published:** 2024-09-24

**Authors:** Hana Kim, Aro Han, Hyunjung Lee, Jiwoo Choi, Hyohjung Lee, Mi-Kyoung Cho

**Affiliations:** 1Department of Nursing, Hoseo University, Asan 31499, Republic of Korea; highting1@hoseo.edu; 2Department of Nursing Science, Research Institute of Nursing Science, School of Medicine, Chungbuk National University, Cheongju 28644, Republic of Korea; adodo23@cbnuh.or.kr (A.H.); 30194@cjhana.com (H.L.); cjw0681@cbnuh.or.kr (J.C.); dlgywjd6711@cbnuh.or.kr (H.L.)

**Keywords:** stroke, self-care, health literacy, digital health, health knowledge, health belief model, self-efficacy

## Abstract

**Background:** The recent substantial increase in the incidence of stroke cases has resulted in high medical expenses. Stroke necessitates ongoing care, emphasizing the importance of consistent self-management. The occurrence of stroke impacts healthcare costs and has far-reaching effects on social services, encompassing disability, unemployment, and other related concerns beyond individuals and families. This study aimed to assess the impact of mobile health literacy, stroke-related health knowledge, health beliefs, and self-efficacy on self-care behaviors of patients with stroke to plan tailored self-care interventions for this patient population. **Methods:** This descriptive survey included 99 stroke patients from three hospitals, which provided treatment equivalent to or better than general hospitals, in City C and was conducted between 7 July 2023 and 30 May 2024. The data collected from hospitalized stroke patients were analyzed using descriptive statistics, independent *t*-tests, one-way ANOVA, and multiple linear regression. **Results:** The self-care behavior of patients with stroke who participated in the study was 73.01 ± 12.24 points. Stroke self-efficacy was identified as a significant factor influencing stroke self-care behaviors and eating habits. Mobile health literacy and stroke self-efficacy also influenced medication and self-care behavior, whereas hypertension and stroke self-efficacy affected lifestyle self-care behaviors. **Conclusions:** Strengthening stroke self-efficacy, improving mobile health literacy, and addressing comorbidities such as hypertension are important for promoting self-care behavior in stroke patients.

## 1. Introduction

Impaired cerebral perfusion from an inadequate blood supply to the brain tissue can lead to stroke with neurological deterioration [1,2]. Approximately 87% of all stroke cases are ischemic strokes [3], and stroke accounts for approximately 6.6 million deaths annually worldwide, making it the second leading cause of death and the foremost reason for acquired disabilities [4]. An increasing number of patients with risk factors for stroke, such as hypertension, hyperlipidemia, obesity, diabetes, and smoking, has resulted in a 70% rise in the incidence of stroke between 1990 and 2019 [5,6]. The recurrence rate has remained stable at 12% after decreasing from 18% before 2005 [7]. Stroke-associated global medical costs exceed USD 891 billion (1.12% of the world’s gross domestic product) [6], and premature death and disability also incur potential losses [8]. South Korea had an annual increase of 1.3% of stroke patients from 2016 to 2020, with the average hospitalization cost per patient reaching KRW 12.57 million [9]. Stroke significantly impacts not only individuals and their families but also healthcare and social welfare services because of the associated medical expenses, disabilities, and job loss.

Stroke often leads to long-term disabilities, resulting in significant life changes and necessitating ongoing self-care behaviors, such as medication, exercise, dietary management, and lifestyle adjustments for treatment and rehabilitation [10]. Stroke has uncontrollable risk factors, such as age, sex, genetics, and race, and controllable risk factors regulated through self-care, such as chronic disease management, smoking, alcohol consumption, and physical activity [1]. Increased self-care behaviors among stroke patients are associated with decreased recurrence and mortality rates [11]. The reduced autonomy of stroke patients in managing their health and the resultant decline in self-care behaviors underscores the relevance of addressing secondary prevention and self-care behaviors by patients and healthcare professionals [12].

A comprehensive knowledge of stroke is crucial for its prevention and management. Understanding the definition, symptoms, management methods, and risk factors of stroke is vital for patients seeking early treatment and preventing recurrence, ultimately leading to lower mortality rates and better outcomes [13]. Patients undergo significant physical, mental, and social changes following stroke and acquire abundant disease-related information [14,15]. Inadequate knowledge of the condition delays the treatment and hinders recovery [15], and lacking an understanding of preventive healthcare measures potentially results in perpetuating risky behaviors [16]. Previous studies have revealed that hospitalized stroke patients often have insufficient information regarding the disease and its prevention [17,18].

The rise of information and communication technology (ICT) has increased the reliance on the Internet and mobile devices for health information [19]. Effectively navigating these resources requires mobile health literacy [19], which entails locating, comprehending, assessing, and utilizing health information on mobile platforms [20]. Studies have shown that higher mobile health literacy is associated with healthier lifestyles and improved self-care, highlighting its growing significance [21]. Thus, enhancing mobile health literacy among stroke patients can empower them to better understand their condition and make well-informed health choices, ultimately reducing the impact of stroke recurrence and improving the overall outcomes [22].

Health beliefs encompass personal values and beliefs about health, including perceived sensitivity, severity, benefits, and barriers [23]. Motivated individuals with a high perception of sensitivity, severity, and benefits of health and a low perception of health barriers are more likely to engage in health behaviors [24]. A high recurrence propensity of stroke encourages patients to adopt health behaviors that support long-term recovery and well-being [25]. Therefore, fostering positive health beliefs in stroke patients can motivate them to embrace beneficial health behaviors [10]. Healthcare professionals can identify factors that influence health behaviors in stroke patients and promote behaviors that enhance health [26].

Self-efficacy pertains to an individual’s confidence in their ability to control and perform the behaviors required to achieve specific tasks [27]. It plays a crucial role in influencing self-care behaviors and eliciting and sustaining human behavioral changes [27]. Enhancing the self-efficacy of stroke patients improves their self-care behaviors and health-promoting practices [28,29]. Self-efficacy is a pivotal factor in maintaining a healthy lifestyle and enhancing the quality of life of stroke patients [30]. This was the most influential factor in encouraging self-care behaviors, indicating that strengthening self-efficacy can improve the performance and continuity of health-related behaviors.

This study has practical implications for stroke patient care and intervention programs. This study aims to delineate the specific impacts of mobile health literacy, stroke-related health knowledge, health beliefs, and self-efficacy on the self-care behaviors of stroke patients. By rigorously analyzing how each of these factors independently and interactively influences self-care, we seek to provide precise, actionable insights that can directly inform the development of targeted self-care intervention programs. Additionally, leveraging our findings, we suggest designing and piloting a mobile-based self-care intervention program tailored specifically to enhance the self-management capabilities of stroke survivors.

## 2. Materials and Methods

### 2.1. Study Design

This descriptive survey was conducted to identify the impact of mobile health literacy, stroke-related health knowledge, health beliefs, and self-efficacy on the self-care behaviors of stroke patients.

### 2.2. Study Population and Sampling

This study involved stroke patients diagnosed and treated at three general hospitals in City C. All outpatients and inpatients were invited to participate in rehabilitation treatment and follow-up observation during the recruitment period. The specific selection criteria were as follows: (1) patients diagnosed with ischemic stroke who had been taking antithrombotic medication for at least one month, (2) patients who were able to communicate and complete a questionnaire, (3) patients who understood the purpose of the study and willingly agreed to participate, (4) adults aged 19 years or older, and (5) patients with first-episode stroke with a National Institutes of Health Stroke Scale (NIHSS) score of 10 or more. The NIHSS score was determined based on Xing and Wei [31], who indicated that patients with a score of 10 or more had the cognitive and physical capacity to complete the self-report questionnaire. Additionally, based on previous research showing differences in self-care behaviors and focus between first-episode and recurrent stroke patients [32], this study focused solely on first-episode patients. Patients diagnosed with hemorrhagic stroke, those hospitalized due to complications from other underlying diseases, those with cognitive impairments who could not understand the questionnaire and perform self-care, and those who did not fully respond to the survey items were excluded from the study.

The required number of study participants was determined using the G*power program version 3.1.9.7, from the Heinrich-Heine-Universität, Düsseldorf, Germany. The sample size was calculated based on a medium effect size of 0.30 [29], with a significance level of 0.05, a statistical power of 0.90, and 16 predictor variables (eight items for participant characteristics, mobile health literacy, stroke-related health knowledge, and health beliefs, such as sensitivity, severity, benefits, barriers, stroke self-efficacy, and stroke self-care behavior). Allowing for a 10% dropout rate, 105 participants were initially selected. This study analyzed 99 of the 105 collected questionnaires after excluding six incomplete responses.

### 2.3. Measurements

#### 2.3.1. Participants Characteristics

The participants’ general characteristics were measured using six items: age, sex, education, economic status, health status, and caregiver status. Disease-related characteristics were measured using two items: stroke duration and number and type of comorbidities for eight items.

#### 2.3.2. Stroke Self-Care Behavior

The assessment of stroke self-care behaviors utilized a tool initially developed by Kang [33], which was later modified and expanded by Kim and Park [34]. The tool comprises 21 items categorized into three subdomains: medication (five items), eating habits (six items), and lifestyle (ten items). Each item was rated on a five-point Likert scale (range: 1–5), with higher scores reflecting better self-care behavior. The tool demonstrated the reliability of Cronbach’s α = 0.81 in Kang’s study [33] and 0.72 in Kim and Park’s study [34]. The tool’s reliability was 0.88 in this study, and the reliability of the subdomains ranged from 0.75 to 0.85.

#### 2.3.3. Mobile Health Literacy

Mobile health literacy was assessed using a tool developed by Norman and Skinner [35] and adapted and translated by Chang et al. [36], which comprises ten items, with eight items contributing to the total score after excluding two items related to health decision-making. Each item is rated on a five-point Likert scale (range: 1–5), with higher scores indicating greater mobile health literacy. The reliability of the tool was Cronbach’s α = 0.88 in Norman and Skinner’s study [35], 0.88 in Chang et al.’s study [36], and 0.96 in this study.

#### 2.3.4. Stroke-Related Health Knowledge

Stroke-related health knowledge was assessed using a tool initially created by Rehe et al. [37] that was later translated into Korean by Chang [38] and adapted by Lee et al. [39]. The tool comprises 25 items rated on a scale of 0 to 1, with higher scores reflecting a greater understanding of stroke-related health knowledge. The tool’s reliability was reported as Cronbach’s α = 0.93 during its development, 0.78 in Lee’s study [39], and 0.76 in this current study.

#### 2.3.5. Health Beliefs

Health beliefs were assessed using a tool based on Becker’s Health Belief Model, originally developed by Byun [40] and adapted for stroke patients by Mun et al. [41]. The tool comprises 20 items, each rated on a five-point Likert scale (ranging from 1 to 5), with higher scores indicating increased sensitivity, severity, benefits, and barriers. The tool demonstrated good reliability, with Cronbach’s α values ranging from 0.67 to 0.83 in the studies by Byun [40] and Mun [41] and from 0.41 to 0.84 in our study.

#### 2.3.6. Stroke Self-Efficacy

Stroke self-efficacy was assessed using a tool developed by Bak [42] and later adapted by Kang and Yoon [33]. This tool comprises 15 items rated on a five-point Likert scale (ranging from 1 to 5), with higher scores indicating greater stroke self-efficacy. The tool demonstrated high reliability, with Cronbach’s α coefficients of 0.86 in Bak’s study [42], 0.84 in Kang and Yoon’s study [33], and 0.89 in this present study.

### 2.4. Data Collection and Ethical Considerations

This study collected data from 99 outpatient and inpatient stroke patients at three general hospitals in City C between 7 July 2023 and 30 May 2024. Approval was obtained from the Institutional Review Board (IRB No. 2023-06-026-001) of C University Hospital before data collection. Participants who provided informed consent after understanding the purpose and content of the study completed a questionnaire. Upon completion, participants submitted the sealed envelope questionnaire directly to the researcher. A small gift (an oral hygiene product) was provided to the participants as a token of appreciation.

### 2.5. Data Analysis

The collected data were analyzed using SPSS for Windows (version 29.0; IBM Corp., Armonk, NY, USA). Descriptive statistics were used to analyze participants’ general characteristics, disease-related characteristics, and variables, including mobile health literacy, stroke-related health knowledge, health beliefs, self-efficacy, and self-care behavior. Based on the participants’ general and disease-related characteristics, independent *t*-tests and one-way analysis ANOVA were used to assess variances in stroke self-care behavior scores were assessed, and the Scheffé test was used for post hoc analysis. Pearson’s correlation coefficient examined the correlation between variables, and multiple linear regression was used to assess the factors influencing stroke self-care behavior.

## 3. Results

### 3.1. Participants’ Characteristics

The average age of the participants was 57.51 ± 11.13 years. This study included more males (*n* = 65, 65.7%) than females and mostly high school graduates (*n* = 51, 51.5%). The average economic status score was 5.18 ± 2.11, with 64 participants (64.6%) below the average economic level. Most caregivers were family members (*n* = 81; 81.8%). The average stroke duration was 3.12 ± 4.10 years, with a disease duration of 1–5 years in the largest group (*n* = 47, 47.4%). The average number of underlying diseases was 1.19 ± 1.04, with hypertension being the most common (*n* = 49, 40.2%), followed by diabetes (*n* = 33, 27.0%) and hyperlipidemia (*n* = 26, 21.3%). The average health status score was 5.17 ± 1.99, with 61 (61.6%) rating their health below average (Table 1).

### 3.2. Measurement Results of the Variables

Stroke self-care behavior among study participants was 73.01 ± 12.24. The scores for the subdomains of stroke self-care behavior were 18.97 ± 3.41 for medication, 19.50 ± 4.11 for eating habits, and 34.55 ± 7.80 for lifestyle. The average mobile health literacy and stroke-related health knowledge were 23.14 ± 7.83 and 19.17 ± 3.72, respectively. The subdomains of health beliefs had the following average scores: sensitivity (16.94 ± 2.54), severity (19.37 ± 3.38), benefits (18.74 ± 3.19), and barriers (16.06 ± 3.72). The average stroke self-efficacy was 54.43 ± 10.03 (Table 2).

### 3.3. Differences in Stroke Self-Care Behavior by Participant Characteristics

The Kolmogorov–Smirnov test confirmed the data normality before analysis. The results indicated statistically significant differences in stroke self-care behavior based on sex and number of comorbidities. Females scored higher than males (*t* = 1.99, *p* = 0.049), and individuals without any comorbidities scored higher than those with two or more comorbidities (F = 4.12, *p* = 0.019). In the medication subdomain of stroke self-care behaviors, individuals without any comorbidities scored significantly higher than those with two or more comorbidities (*t* = 4.57, *p* = 0.013). In the eating habits subdomain, females scored higher than males (*t* = 2.89, *p* = 0.005), and individuals with above-average economic status scored higher than those with below-average economic status (*t* = −2.39, *p* = 0.019). Finally, statistically significant differences were observed based on the presence of hypertension in the lifestyle subdomain, with non-hypertensive individuals scoring higher than hypertensive individuals (*t* = −3.11, *p* = 0.002) (Table 3).

### 3.4. Correlations among the Variables

This study revealed significant positive correlations between stroke self-care behavior and stroke-related health knowledge (*r* = 0.21, *p* = 0.037), as well as health beliefs as benefits (*r* = 0.34, *p* < 0.001) and stroke self-efficacy (*r* = 0.64, *p* < 0.001). Furthermore, the medication aspect of stroke self-care behavior correlated positively with mobile health literacy (*r* = 0.36, *p* < 0.001), stroke-related health knowledge (r = 0.29, *p* = 0.004), benefits (*r* = 0.44, *p* < 0.001), and stroke self-efficacy (*r* = 0.55, *p* < 0.001). Additionally, the eating habits subdomain was positively correlated with benefits (*r* = 0.29, *p* = 0.004) and stroke self-efficacy (*r* = 0.50, *p* < 0.001). Finally, the lifestyle subdomain correlated positively with benefits (*r* = 0.22, *p* = 0.029) and stroke self-efficacy (*r* = 0.52, *p* < 0.001) (Table 4).

### 3.5. Factors Influencing Stroke Self-Care Behavior

An analysis utilizing multiple linear regression models was used to investigate the determinants of stroke self-care behaviors encompassing various predictive factors across four distinct models: total self-care, medication adherence, eating habits, and lifestyle management (Table 5).

The total self-care behavior model, including variables, such as sex, number of comorbidities, stroke-related health knowledge, perceived benefits, and stroke self-efficacy, explained 40.4% of the variance (F = 66.76, *p* < 0.001). Stroke self-efficacy was a significant predictor in this model.

In the medication adherence model, the variables considered were the number of comorbid diseases, mobile health literacy, stroke-related health knowledge, perceived benefits, and stroke self-efficacy, which explained 29.2% of the variance (F = 21.02, *p* < 0.001). Mobile health literacy and stroke self-efficacy were identified as key factors influencing medication self-care behaviors.

The eating habits model included factors such as sex, economic status, perceived benefits, and stroke self-efficacy, which accounted for 24.7% of the variability (F = 33.07, *p* < 0.001). This study identified stroke self-efficacy as a crucial determinant.

Finally, the lifestyle management model, considering the presence of hypertension, perceived benefits, and stroke self-efficacy, explained 31.2% variability (F = 22.12, *p* < 0.001). The presence of hypertension and higher stroke self-efficacy was associated with improved lifestyle self-care behaviors.

## 4. Discussion

This research aimed to determine the factors influencing stroke patients’ self-care behavior. The findings revealed that stroke self-efficacy significantly impacted overall self-care behavior. Furthermore, mobile health literacy and stroke self-efficacy played a vital role in medication self-care. Regarding dietary habits and lifestyle, stroke self-efficacy emerged as the primary influencing factor. The presence of hypertension, in conjunction with stroke self-efficacy, also affected lifestyle-related self-care behavior.

The study participants showed an average self-care behavior score of 73.01 ± 12.24 out of 100, which was lower than the score reported (81.67 ± 14.72) by Park et al. [43] using the same tool. Park et al.’s [43] higher scores could be attributed to the greater self-care autonomy of the living environment of their outpatient participants. Simultaneously, our study included both outpatients and inpatients. However, the stroke self-care behavior score (75.6 ± 13.44) in an inpatient-based study by Ryu et al. [10] was higher than that in our study. This disparity may be due to the difference in average stroke duration (2.34 years in Ryu et al.’s study [10] vs. 3.12 years in our study), reflecting the tendency for a higher stroke self-care behavior with a shorter stroke duration [44].

The study participants were chosen based on a NIHSS score of 10 or more, as indicated by Xing and Wei [31], who found that patients within this range possessed the cognitive and physical capacity necessary to complete the self-report questionnaire. This study specifically focused on patients experiencing their first stroke episode, taking into account the differing self-care behaviors and focusing on first-episode and recurrent stroke patients [32]. Saber and Saver [45] reported that approximately 71% of stroke hospitalizations in the United States are attributed to patients with an NIHSS score of 0–4, while only 29% are attributed to those with a score of 10 or higher. Despite the small sample size and convenience sampling, the participants of this study can be considered representative of stroke patients. Therefore, the factors influencing self-care behavior identified in this study can be deemed sufficiently viable as foundational data for developing interventions for stroke patients.

In a recent study, stroke self-efficacy influenced stroke self-care behaviors significantly. Specifically, higher levels of stroke self-efficacy were associated with better performance in stroke self-care behaviors, such as medication management, eating habits, and lifestyle maintenance. This suggests that stroke patients with greater stroke self-efficacy are more likely to achieve physical rehabilitation, minimize disability, and reintegrate into social life through proactive and consistent self-care, starting from early disease stages [29,46]. Self-efficacy, an individual’s perceived confidence in managing chronic diseases, is crucial for promoting a healthy lifestyle, improving quality of life [30], and enabling behavioral changes [47,48]. These findings align with previous studies [10,29,48,49], underscoring the significance of self-efficacy in fostering self-care behavior among stroke patients. These results make it imperative to assess stroke self-efficacy levels in specific self-care domains when providing interventions for stroke patients. Tailored interventions to foster their belief in the successful management of doubtful tasks could substantially contribute to their long-term adaptation to disease-posed challenges.

This study found that mobile health literacy significantly influenced self-care behavior regarding medication. Enhanced mobile health literacy was linked to improved self-care behavior regarding medication. This supports the findings of Park et al. [43], who observed that higher e-health literacy, a mobile health literacy-related concept, was associated with greater health interest, health information expectations, and proactive health behavior. Additionally, Wenjing et al. [50] reported the association between the reduced understanding of prescribed medication guidelines and lower health literacy, leading to non-adherence. Kim et al. [51] discovered that a smartphone-based management system improved medication adherence, stroke awareness, depression alleviation, and quality of life. They also noted that mobile technology-enhanced medication adherence, specifically through a self-management intervention using WeChat [51]. A systematic review of mobile app-based interventions for patients with cardiovascular disease confirmed their effectiveness in improving medication self-care behavior [52]. Therefore, incorporating elements that improve mobile health literacy in mobile healthcare services may effectively enhance medication self-care behaviors among stroke patients. This approach could also improve post-discharge rehabilitation accessibility through remote rehabilitation with comparable effectiveness to traditional treatments and potentially reduce medical costs [53,54]. This study did not adequately account for health literacy, digital resource accessibility, and socioeconomic factors, which suggests that the impact of mobile health literacy on medication self-care behaviors may have been overestimated. Therefore, it is essential to exercise caution when interpreting the results. However, it is worth noting that in Korea, health literacy and digital resource accessibility are excellent. The country ensures medical accessibility through National Health Insurance and medical protection programs, allowing people to access health information easily [55]. With an illiteracy rate of less than 1% [56], the overall health literacy in Korea is high. Moreover, Korea boasts one of the world’s best Internet infrastructures [57], and 93.0% of the population uses smartphones [58], indicating a robust digital infrastructure. Nevertheless, previous research has shown that individuals with high health literacy tend to exhibit better self-care behaviors [59]. In contrast, those with low mobile literacy may struggle to utilize mobile-based services effectively [21]. Sieck et al. [21] highlighted how limitations in digital accessibility can have a negative impact on self-care behavior, and Smith and Magnani [60] emphasized the significant influence of socioeconomic factors on patients’ ability to use digital devices and access medical services. Therefore, it is essential to approach the findings of this study with caution, and future research should consider these factors in its design.

This study found that hypertension significantly influenced lifestyle self-care behavior. Patients with hypertension showed better stroke behaviors than those without hypertension. According to Kao et al. [61], comorbidities, including hypertension, significantly affect self-care and functional recovery in stroke patients. Similarly, Wardhani [62] reported that hypertension, diabetes, and hyperlipidemia negatively impact self-care and NIHSS score improvements. Hypertension was the most common comorbidity among participants (40.2%), consistent with previous studies [49,63,64]. It is a significant risk factor for stroke, affecting approximately 70% of stroke patients [65]. This condition increases stress on the vascular walls, causes endothelial dysfunction, stiffens large arteries in the brain, and contributes to atherosclerosis and carotid artery disease, thereby increasing the likelihood of stroke [66]. Approximately 20% of all ischemic stroke cases occur in patients with hypertension [65]. Living with hypertension also leads to greater awareness and knowledge of the condition, resulting in better self-care behaviors [67]. This study found that the average stroke duration among the subjects was 3.12 ± 4.10 years. This indicates that individuals may be more proactive in recognizing and managing their highest risk during a relatively short timeframe. Additionally, the management of hypertension is considered to be more straightforward compared to diabetes or hyperlipidemia, possibly because patients can more easily address it through medication and lifestyle adjustments [68,69]. Additionally, lifestyle modification and management effectively prevent stroke compared to other methods that treat stroke risk factors [22,48]. However, some stroke survivors mistakenly believe that taking preventive medication is more essential than modifying lifestyle behaviors [70]. In spite of this, medication nonadherence is high among stroke patients [71]. Thus, lifestyle management plays a critical role in stroke prevention and recurrence [48]. Therefore, improving stroke patients’ self-care abilities requires a comprehensive approach that emphasizes the need for lifestyle changes along with other self-care areas.

Contrary to previous research, the regression analysis in this study did not find a significant relationship between specific participant characteristics and self-care behavior. Sex was not identified as a substantial factor influencing self-care behavior, aligning with findings from Sun et al. [72], Delgado et al. [73], and Dawson et al. [74]. However, Jirjees et al. [75] revealed that women had a higher level of stroke knowledge than men, which could impact self-care behavior. Venditti et al. [76] and Arapi et al. [77] reported conflicting results, suggesting that men had higher self-care behaviors. These discrepancies may be due to differences in research methods, target groups, and cultural influences, highlighting the need for further investigation into gender differences. While previous research indicated that patients with comorbidities had lower self-care behavior levels [72], our study found that the number of comorbidities did not significantly affect self-care behavior. This result aligns with studies by Park and Kim [63] and Kuo et al. [78]. Notably, the presence of hypertension was found to influence stroke self-care behaviors, suggesting the need for a nuanced approach considering the type, severity, and frequency of diseases rather than just the number of comorbidities. Stroke-related health knowledge was also not identified as a significant factor influencing self-care behavior. While patients with increased health education and knowledge may exhibit better self-care behavior [79], the study did not compare the disease-related education experience before and after stroke diagnosis. This limited the confirmation of the impact of knowledge on self-care behavior. Additionally, although 70.8% of the study participants had general knowledge about stroke, only about 20% knew all the symptoms and risk factors [75]. The average stroke-related health knowledge score was 19.17 out of 25 points, but it is important to note that the overall knowledge of stroke was measured using only 25 questions. Furthermore, 36.4% of the participants had experienced a stroke in less than one year, indicating that the focus was mainly on acute treatment, and the overall level of knowledge about stroke was low, which did not significantly affect self-care behavior. This study found that health beliefs did not considerably impact self-care behavior, which aligns with a study by Ryu et al. [10], which focused on patients admitted to nursing hospitals. It was noted that the patients’ health beliefs may not have influenced their self-care behavior, possibly due to the limited scope of their actions. This study included recently admitted patients, which could partially explain the results. Additionally, it was observed that behavioral patterns and health beliefs in patients with chronic diseases begin to take shape within a few months after diagnosis and progress steadily [80]. The average stroke duration in the two studies’ participants was 2.34 years in Ryu et al. [10] and 3.12 years in this study. In addition, 36.4% (n = 36) of patients had a stroke within one year of onset, and 83.8% (n = 83) of patients had a stroke less than five years of onset, which means that many subjects were in the early stages of the disease. It can be concluded that this was a period when health beliefs about stroke were not sufficiently formed.

In this study, we conducted a comprehensive review of previous studies and incorporated variables deemed to influence the self-care behavior of stroke patients. We assessed the impact of these variables on self-care; however, it is critical to note that not all potential influencing factors were considered, nor were the interrelationships between self-care and other variables thoroughly explored. Wang et al. [81] discovered that not only did stroke patients’ self-care improve, but caregivers’ proactive attitudes also had a positive impact. Additionally, Peyman [82] affirmed that clear communication and easily understandable educational materials enhanced patients’ self-care behavior. Moreover, Shuqi et al. [79] and Chau et al. [83] found that improved access to medical care was associated with greater self-efficacy and self-care behavior. Conversely, Babkair and Dickson [84] established that low socioeconomic status and a lack of family and social support were limiting factors for self-care behavior. In Korea, family members traditionally care for patients due to the country’s medical system and culture [85]. Additionally, the National Health Insurance Service covers 97% of the population, ensuring high accessibility to medical care for all citizens [55]. However, the significant out-of-pocket expenses (approximately 65%) may impede medical accessibility for low-income people [55]. Future studies should consider analyzing variables related to caregivers’ attitudes, stroke education, social status, medical care accessibility, and support from family and society to strengthen the conclusions of this study.

This study has several limitations. Firstly, the participants were convenience-sampled, leading to selection bias. For instance, patients who were more motivated to manage their health or had better access to healthcare facilities may have participated in our sample. Consequently, our findings may not fully represent the broader stroke patient population, limiting the generalizability of the results. Secondly, the primary variables in this study were gathered through self-report questionnaires, which are instrumental for assessing subjective aspects of self-care behaviors. However, self-report data might be overestimated due to limitations such as memory errors and social desirability or recall biases. To address these issues, future studies could benefit from incorporating objective data sources that provide verifiable evidence of self-care behaviors. These could include physiological indicators such as blood pressure, blood sugar, and blood cholesterol levels, as well as medication adherence and diet records validated by monitoring devices like intelligent pill bottles [86]. Cross-referencing these objective measures with self-report questionnaires would help ensure the accuracy and reliability of the findings. Thirdly, the mobile health literacy tool used in this study was validated for Korean adults but not stroke patients, and the other tools’ validity still needs to be confirmed. Previous studies on domestic stroke patients used tools other than mobile health literacy. Although expert validity was assessed during this study, establishing the tool’s validity may not be sufficient. Therefore, future research should utilize tools that have been validated for domestic stroke patients. Additionally, the sensitivity reliability was low at 0.41 when measuring health beliefs in this study. This could be attributed to the tool’s limited number of items or the survey being conducted during outpatient treatment and hospitalization for first-episode stroke patients [87], reflecting the participants’ emotional distress and potential misunderstanding of the survey items [88,89]. To address this, future studies could consider utilizing a mobile survey method so that patients can respond in a more stable environment after returning home. Fourthly, this study had limitations, including a small sample size and restricted geographical representation, which may impact the generalizability of our findings. Future research should consider demographic factors such as age, gender, income, and education level and use methods like proportional stratified random sampling to enhance the applicability of the results. Additionally, using a cross-sectional study design limits the establishment of causal relationships due to the lack of a time dimension [90], lowering the level of evidence compared to research designs like randomized controlled trials. Therefore, future studies should explore factors influencing self-care using longitudinal research designs such as randomized controlled trials and cohort studies.

## 5. Conclusions

This study analyzed the impact of mobile health literacy, stroke-related health knowledge, health beliefs, and self-efficacy on stroke self-care behavior. We surveyed 99 stroke patients, both inpatients and outpatients, across three general hospitals in City C. The results showed the significant influence of stroke-related self-efficacy on stroke self-care behaviors. Moreover, this study revealed that mobile health literacy and stroke self-efficacy significantly affected self-care regarding medication among stroke patients. Stroke self-efficacy was identified as a key influencing factor for eating habits and lifestyle. Additionally, hypertension and stroke self-efficacy affected lifestyle aspects of self-care behavior among stroke patients. Overall, this study suggests the development of comprehensive, individualized self-care programs that integrate health education, technological tools, and psychosocial support to enhance stroke self-care behaviors and optimize patient outcomes. Continuous management is crucial for patients with conditions like stroke. Developing a customized mobile app could help ensure ongoing rehabilitation and self-care behaviors. This approach would be particularly suitable in Korea, given its well-established mobile service environment and high smartphone usage. Moreover, by providing the service through hospitals, we can anticipate improved health management effectiveness and reduced medical costs by using health data beyond hospital settings.

## Figures and Tables

**Table 1 healthcare-12-01913-t001:** Descriptive statistics of participant characteristics (N = 99).

Characteristics	Categories	*n* (%)	M ± SD	Min~Max
Age (years)	<57.51	42 (42.4)	57.51 ± 11.13	22.00~79.00
	≥57.51	57 (57.6)		
Gender	Female	34 (34.3)		
	Male	65 (65.7)		
Education	≤Middle school	21 (21.2)		
	High school	51 (51.5)		
	≥College	27 (27.3)		
Economic status	<5.18	64 (64.6)	5.18 ± 2.11	0.00~10.00
	≥5.18	35 (35.4)		
Care-giver	Family	81 (81.8)		
	Non-family	8 (8.1)		
	None	10 (10.1)		
Duration of stroke (years)	<1	36 (36.4)	3.12 ± 4.10	0.17~26.17
	≥1 and <5	47 (47.4)		
	≥5	16 (16.2)		
Comorbidity ^†^ (number)	None	31 (31.3)	1.19 ± 1.04	0.00~4.00
	1	31 (31.3)		
	≥2	37 (37.4)		
Comorbidity ^†^ (type)	Hypertension	49 (40.2)		
	Diabetes mellitus	33 (27.0)		
	Hyperlipidemia	26 (21.3)		
	Heart disease	9 (7.4)		
	Others	5 (4.1)		
Health status	<5.17	61 (61.6)	5.17 ± 1.99	0.00~10.00
	≥5.17	38 (38.4)		

Notes: M, mean; SD, standard deviation; Min, minimum; Max, maximum. ^†^ Multiple response.

**Table 2 healthcare-12-01913-t002:** Descriptive Statistics of the Variables (N = 99).

Variables	Items	M ± SD	Min–Max	Scale Standardized Score	
				M ± SD	Min~Max
Stroke self-care behavior	21	73.01 ± 12.24	49.00~99.00	3.48 ± 0.58	2.33~4.71
Medication	5	18.97 ± 3.41	12.00~25.00	3.79 ± 0.68	2.40~5.00
Eating Habits	6	19.50 ± 4.11	10.00~30.00	3.25 ± 0.68	1.67~5.00
Lifestyle	10	34.55 ± 7.80	10.00~50.00	3.45 ± 0.78	1.00~5.00
Mobile health literacy	8	23.14 ± 7.83	8.00~40.00	2.89 ± 0.98	1.00~5.00
Stroke-related health knowledge	25	19.17 ± 3.72	8.00~25.00	0.77 ± 0.15	0.32~1.00
Health beliefs					
Sensitivity	5	16.94 ± 2.54	10.00~23.00	3.39 ± 0.59	2.00~5.00
Severity	5	19.37 ± 3.38	10.00~25.00	3.87 ± 0.68	2.00~5.00
Benefit	5	18.74 ± 3.19	10.00~25.00	3.75 ± 0.64	1.67~5.00
Barrier	5	16.06 ± 3.72	7.00~23.00	3.21 ± 0.74	1.00~5.00
Stroke self-efficacy	15	54.43 ± 10.03	37.00~75.00	3.63 ± 0.67	2.47~5.00

Notes: M, mean; SD, standard deviation; Min, minimum; Max, maximum.

**Table 3 healthcare-12-01913-t003:** The differences in stroke self-care behavior according to participant characteristics (N = 99).

Characteristics	Categories	Stroke Self-Care Behavior							
		Total		Medication		Eating Habits		Lifestyle	
		M ± SD	*t* or F (*p*) Scheffé	M ± SD	*t* or F (*p*) Scheffé	M ± SD	*t* or F (*p*) Scheffé	M ± SD	*t* or F (*p*) Scheffé
Age (year)	<57.51	72.68 ± 11.87	−0.36 (0.715)	18.66 ± 3.29	−1.03 (0.304)	19.33 ± 3.54	−0.45 (0.651)	34.68 ± 7.50	−0.36 (0.715)
	≥57.51	73.35 ± 12.90		19.39 ± 3.59		19.71 ± 4.80		34.35 ± 8.26	
Sex	Female	76.41 ± 11.58	1.99 (0.049)	19.64 ± 3.47	1.43 (0.154)	21.08 ± 3.53	2.89 (0.005)	35.67 ± 7.92	1.04 (0.299)
	Male	71.16 ± 12.30		18.60 ± 3.36		18.66 ± 4.16		33.95 ± 7.72	
Education	≤Middle school	71.00 ± 10.52	0.93 (0.397)	18.38 ± 3.49	1.57 (0.213)	19.71 ± 3.53	0.03 (0.963)	32.90 ± 8.11	1.14 (0.323)
	High school	72.47 ± 12.81		18.70 ± 3.27		19.45 ± 4.26		34.31 ± 7.95	
	≥College	75.59 ± 12.37		19.96 ± 3.57		19.40 ± 4.36		36.25 ± 7.18	
Economic status	<5.18	71.39 ± 11.57	−1.83 (0.070)	18.69 ± 3.48	−1.05 (0.296)	18.78 ± 3.59	−2.39 (0.019)	33.84 ± 7.80	−1.21 (0.228)
	≥5.18	76.08 ± 13.10		19.45 ± 3.30		20.80 ± 4.68		35.82 ± 7.73	
Caregiver	Family	73.06 ± 12.09	2.15 (0.122)	18.91 ± 3.50	0.05 (0.942)	19.71 ± 3.88	2.41 (0.094)	34.35 ± 7.70	2.71 (0.070)
	Non- family	79.75 ± 14.78		19.25 ± 3.49		20.50 ± 5.52		40.00 ± 8.96	
	None	67.8 ± 10.05		19.20 ± 2.93		16.90 ± 4.17		31.70 ± 6.03	
Duration of stroke (year)	<1	71.72 ± 12.91	0.84 (0.431)	19.22 ± 3.69	1.21 (0.300)	19.08 ± 4.25	0.28 (0.751)	33.41 ± 8.48	0.92 (0.400)
	≥1 and <5	74.78 ± 11.73		19.19 ± 3.31		19.76 ± 4.01		35.65 ± 7.24	
	≥5	71.18 ± 12.50		17.75 ± 3.02		19.62 ± 4.24		33.81 ± 7.78	
Comorbidity (number)	None ^a^	76.70 ± 11.62	4.12 (0.019) a > c	20.43 ± 3.12	4.57 (0.013) a > c	20.70 ± 3.76	2.63 (0.077)	35.29 ± 8.13	2.78 (0.067)
	1 ^b^	68.22 ± 11.67		19.96 ± 2.98		18.35 ± 4.11		31.90 ± 7.82	
	≥2 ^c^	74.18 ± 12.31		18.62 ± 3.67		19.43 ± 4.20		36.13 ± 7.09	
Hypertension	Yes	72.93 ± 12.61	−0.10 (0.916)	19.26 ± 4.00	−0.54 (0.585)	35.73 ± 7.47	1.51 (0.134)	17.93 ± 3.64	−3.11 (0.002)
	No	73.20 ± 12.07		19.72 ± 4.23		33.38 ± 8.00		20.00 ± 2.86	
Diabetes mellitus	Yes	73.51 ± 13.10	0.25 (0.080)	19.12 ± 4.43	−0.62 (0.536)	35.72 ± 7.24	0.63 (0.525)	18.66 ± 3.85	1.06 (0.289)
	No	72.84 ± 11.94		19.61 ± 3.95		33.72 ± 7.24		19.12 ± 3.20	
Hyperlipidemia	Yes	71.96 ± 11.08	−0.53 (0.594)	18.88 ± 3.14	−0.14 (0.884)	18.96 ± 4.01	−0.76 (0.444)	34.11 ± 7.46	−0.32 (0.745)
	No	73.47 ± 12.73		19.00 ± 3.54		19.68 ± 4.15		34.69 ± 7.95	
Heart disease	Yes	72.33 ± 15.45	−0.18 (0.851)	19.88 ± 2.75	0.84 (0.401)	19.88 ± 4.31	0.30 (0.765)	32.55 ± 10.00	−0.80 (0.425)
	No	73.14 ± 12.02		18.87 ± 3.48		19.45 ± 4.11		34.74 ± 7.58	
Health status	<5.17	72.77 ± 12.35	−0.31 (0.757)	18.81 ± 3.25	−0.55 (0.581)	19.36 ± 4.05	−0.41 (0.683)	34.59 ± 7.72	0.55 (0.581)
	≥5.17	73.56 ± 12.32		19.21 ± 3.71		19.71 ± 4.24		34.47 ± 8.02	

Notes: M, mean; SD, standard deviation. ^a, b, c^ comparison groups of Scheffe test.

**Table 4 healthcare-12-01913-t004:** Correlation among the variables.

Variables	1	1.1	1.2	1.3	2	3	4.1	4.2	4.3	4.4	5
	*r* (*p*)										
1. Stroke self-care behavior	1										
1.1 Medication	0.29 (0.004)	1									
1.2 Diet habit	0.13 (0.215)	0.54 (<0.001)	1								
1.3 Lifestyle	0.15 (0.130)	0.35 (<0.001)	0.58 (<0.001)	1							
2. Mobile health literacy	0.11 (0.297)	0.36 (<0.001)	0.07 (0.501)	−0.01 (0.899)	1						
3. Stroke-related health knowledge	0.21 (0.037)	0.29 (0.004)	0.13 (0.215)	0.15 (0.130)	0.39 (<0.001)	1					
4.1 Health belief (sensitivity)	0.10 (0.347)	0.07 (0.493)	0.05 (0.631)	0.09 (0.397)	−0.03 (0.739)	0.08 (0.458)	1				
4.2 Health belief (severity)	0.07 (0.493)	0.19 (0.060)	0.03 (0.748)	0.02 (0.829)	−0.02 (0.069)	−0.09 (0.380)	0.34 (<0.001)	1			
4.3 Health belief (benefit)	0.34 (<0.001)	0.44 (<0.001)	0.29 (0.004)	0.22 (0.029)	0.24 (0.018)	0.23 (0.022)	0.13 (0.204)	0.17 (0.100)	1		
4.4 Health belief (barrier)	−0.04 (0.691)	−0.06 (0.577)	−0.20 (0.855)	−0.03 (0.787)	−0.018 (0.070)	−0.10 (0.342)	0.16 (0.109)	0.44 (<0.001)	0.13 (0.185)	1	
5. Stroke self-efficacy	0.64 (<0.001)	0.55 (<0.001)	0.50 (<0.001)	0.52 (<0.001)	0.32 (0.001)	0.33 (0.001)	0.05 (0.594)	0.18 (0.072	0.61 (<0.001)	−0.10 (0.342)	1

Note: r: Pearson’s correlation coefficient.

**Table 5 healthcare-12-01913-t005:** Factors affecting the stroke self-care behavior.

Variables	Stroke Self-Care Behavior											
	Total			Medication			Eating Habit			Lifestyle		
	B (S.E.)	*β*	*t* (*p*)	B (S.E.)	*β*	*t* (*p*)	B(S.E.)	*β*	*t* (*p*)	B (S.E.)	*β*	*t* (*p*)
(Constant)	30.35 (5.32)		5.71 (<0.001)	8.61 (1.66)		5.77 (<0.001)	8.26 (1.99)		4.16 (<0.001)	12.87 (3.61)		3.57 (<0.001)
Hypertension (ref. = none)										3.60 (1.32)	0.23	2.74 (0.007)
Mobile health literacy				0.11 (0.04)	0.24	2.70 (0.008)						
Stroke self -efficacy	0.78 (0.10)	0.64	8.17 (<0.001)	0.15 (0.03)	0.43	4.72 (<0.001)	0.21 (0.04)	0.50	5.75 (<0.001)	0.43 (0.07)	0.56	6.56 (<0.001)
F (*p*)	66.76 (<0.001)			21.02 (<0.001)			33.07 (<0.001)			22.12 (<0.001)		
Adj. R^2^ (%)	40.4			29.2			24.7			31.2		
Tolerance	1.00			0.90			1.00			0.98		
VIF	1.00			1.11			1.00			1.02		
Durbin–Watson	1.90			1.58			1.70			2.05		

Notes: B, unstandardized coefficients; SE, standard error; VIF, variance inflation factors.

## Data Availability

The original contributions presented in the study are included in the article, further inquiries can be directed to the corresponding authors.

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
