# Peer review of "Impact of Mobile Health Literacy, Stroke-Related Health Knowledge, Health Beliefs, and Self-Efficacy on the Self-Care Behavior of Patients with Stroke"

_healthcare, 2024, doi:10.3390/healthcare12191913_

Round 1

Reviewer 1 Report

Comments and Suggestions for Authors

Dear Authors,

Thank you for submitting your manuscript. I have identified some areas where you might consider/clarify. Below are the detailed suggestions and comments:

1. Sample Size and Generalizability

The study has a relatively small sample size of 99 participants, which limits the generalizability of the findings, particularly to populations outside the studied region.

Please expand the discussion to further acknowledge this limitation and clarify how the findings may be applicable (or not) to other healthcare settings or populations. Additionally, if feasible, future studies should aim to include a larger and more diverse sample.

2. Cross-Sectional Design

The cross-sectional design limits the ability to establish causal relationships between variables, such as mobile health literacy and self-care behavior.

 Please explicitly discuss this limitation in the manuscript. A recommendation for future research could be to conduct longitudinal studies to better understand how these relationships evolve over time.

3. Selection Bias

The use of convenience sampling increases the risk of selection bias, as the sample may over-represent certain characteristics, such as patients with higher health motivation or access to care.

 Please include a section discussing potential selection bias and its impact on the findings. Acknowledging that the sample may not be representative of the broader stroke population would strengthen the manuscript.

4. Overreliance on Self-Reported Data

The reliance on self-reported data may introduce biases such as social desirability or recall bias, especially in measures related to self-care behaviors and health literacy.

Please discuss the limitations of self-reported data more thoroughly. If possible, consider suggesting alternative data collection methods, such as cross-verifying self-reported behaviors with clinical records (e.g., medication adherence) in future studies.

5. Comorbidity and Health Status Data

While the manuscript includes comorbidity and health status data, the impact of these factors on the different domains of self-care behavior (medication, eating habits, lifestyle) is not fully explored.

 Please include a more detailed analysis of how comorbidities, such as diabetes or hypertension, affect stroke self-care behaviors. This will provide a more nuanced understanding of the factors influencing self-care beyond self-efficacy and health literacy.

6. Confounding Variables

The study does not adequately account for potential confounding variables such as socio-economic status, caregiver support, or healthcare access, which could affect both self-efficacy and self-care behaviors.

Please include a discussion of potential confounding variables and how they may have influenced the results. A more robust multivariate analysis that accounts for these factors could strengthen your conclusions.

7. Measurement Tools

Some of the tools used to measure variables, particularly health beliefs, have lower reliability scores (Cronbach’s α = 0.41 to 0.84), raising concerns about consistency.

 Please discuss the reliability limitations of the measurement tools and consider recommending alternative or additional validated instruments in future research.

8. Detailed Intervention Suggestions

While the study provides useful findings, the manuscript lacks practical recommendations on how these insights can be translated into clinical or public health interventions.

 Please expand the discussion to include actionable recommendations for healthcare professionals and policymakers. For example, you could suggest specific mobile health interventions or educational programs that target health literacy and self-efficacy.

9. Underreporting of Negative or Non-Significant Results

 The manuscript focuses on positive relationships between variables, such as self-efficacy and self-care behaviors, but it does not adequately address non-significant findings.

Please include a discussion of variables that did not show significant relationships and explore potential reasons for these findings. This will provide a more balanced interpretation of the results. 10. Over-Estimation of Stroke Self-Efficacy Impact

The manuscript may over-estimate the influence of stroke self-efficacy, without fully considering other contextual factors like family support or access to healthcare.

Please include a more balanced interpretation by acknowledging that while self-efficacy is important, other external factors could significantly affect self-care behaviors. A more nuanced discussion of these factors will provide a deeper understanding of stroke management.

11. Mobile Health Literacy Over-Estimation

The impact of mobile health literacy on medication adherence may be over-estimated, given that general health literacy and access to digital resources are not fully addressed.

 Consider expanding the discussion to explore how general health literacy, digital accessibility, and socio-economic factors may moderate the relationship between mobile health literacy and self-care behavior.

 looking forward to seeing your revised manuscript.

Best wishes,

Comments on the Quality of English Language

Consider using more precise or varied vocabulary to avoid repetition. For example, "important" could be replaced with "essential" or "critical" in some places to add variety to the text.

Avoid repetition of phrases like "the study was conducted" and "the survey was conducted." These could be consolidated for conciseness, e.g., "The descriptive survey of stroke patients was conducted between July 7, 2023, and May 30, 2024."

Author Response

Response to Reviewer 1 Comments

We appreciate your valuable feedback on this aspect. Additionally, we acknowledge the need for further refinement to enhance the clarity and contribution of our study. We will consider your suggestions to improve our research's interpretability and relevance. By implementing these revisions, we aim to ensure that our study is more accessible and meaningful so readers can comprehend the research content and findings. The revised parts of the text are indicated in blue font.

Point 1. Sample Size and Generalizability

The study has a relatively small sample size of 99 participants, which limits the generalizability of the findings, particularly to populations outside the studied region. Please expand the discussion to further acknowledge this limitation and clarify how the findings may be applicable (or not) to other. Additionally, if feasible, future studies should aim to include a larger and more diverse sample.

Response 1: Thank you for your valuable feedback regarding our study's sample size and generalizability. In recognition of your concerns, we have expanded our discussion of these limitations in the manuscript. Specifically, we explicitly addressed the constraints posed by our sample size and geographical representation, which might impact the generalizability of our findings. Additionally, we have outlined potential methodological improvements for future research, such as incorporating diverse demographic factors and employing proportional stratified random sampling, to enhance the robustness and applicability of the results across different populations.

Page 13, lines 460-469

Fourthly, the study had limitations, including a small sample size and restricted geographical representation, which may impact the generalizability of our findings. Future research should consider demographic factors such as age, gender, income, and education level and use methods like proportional stratified random sampling to enhance the applicability of the results.

Point 2. Cross-Sectional Design

The cross-sectional design limits the ability to establish causal relationships between variables, such as mobile health literacy and self-care behavior.  Please explicitly discuss this limitation in the manuscript. A recommendation for future research could be to conduct longitudinal studies to better understand how these relationships evolve over time.

Response 2: Thank you for your thorough review and constructive comments on our study's design. We appreciate your insights regarding the limitations inherent to the cross-sectional approach used in our research. In response to your feedback, we have revised the manuscript to provide a more detailed discussion of these limitations, specifically the inability to establish causal relationships due to the lack of a time dimension. Furthermore, we have recommended future research methodologies that could overcome these limitations. We suggest that subsequent studies could employ longitudinal designs, such as randomized controlled trials and cohort studies, to understand and possibly establish the causal relationships between mobile health literacy and self-care behaviors over time.

Page 13, lines 464-469

Additionally, using a cross-sectional study design limits the establishment of causal relationships due to the lack of a time dimension [90], lowering the level of evidence compared to research designs like randomized controlled trials. Therefore, future studies should explore factors influencing self-care using longitudinal research designs such as randomized controlled trials and cohort studies.

Point 3. Selection Bias

The use of convenience sampling increases the risk of selection bias, as the sample may over-represent certain characteristics, such as patients with higher health motivation or access to care. Please include a section discussing potential selection bias and its impact on the findings. Acknowledging that the sample may not be representative of the broader stroke population would strengthen the manuscript.

Response 3: We appreciate your thoughtful comment regarding the potential selection bias introduced by the use of convenience sampling in our study. We explicitly acknowledged that the sampling method may have resulted in an overrepresentation of certain patient characteristics, such as higher health motivation and access to care, which could influence the findings. Furthermore, we recognize that our sample may not be fully representative of the broader stroke patient population. Based on your advice, we have included a more detailed discussion of this limitation in the manuscript.

Page 13, lines 435-439

The study has several limitations. Firstly, the participants were convenience sampling leading to selection bias. For instance, patients who were more motivated to manage their health or had better access to healthcare facilities may have participated in our sample. Consequently, our findings may not fully represent the broader stroke patient population, limiting the generalizability of the results.

Point 4. Overreliance on Self-Reported Data

The reliance on self-reported data may introduce biases such as social desirability or recall bias, especially in measures related to self-care behaviors and health literacy. Please discuss the limitations of self-reported data more thoroughly. If possible, consider suggesting alternative data collection methods, such as cross-verifying self-reported behaviors with clinical records (e.g., medication adherence) in future studies.

Response 4: Thank you for your meticulous review and the valuable insights regarding the reliability of self-reported data. We have taken your advice to heart and expanded our discussion on the potential biases introduced by this data collection method, such as social desirability and recall biases, particularly concerning self-care behaviors and health literacy. Furthermore, we have included suggestions for future research approaches that could mitigate these limitations. Specifically, we propose the integration of objective data sources, such as physiological measures and medication/diet adherence records, which can be cross-verified with self-reported behaviors to enhance the validity of the findings.

Page 13, lines 439-448

Secondly, the primary variables in this study were gathered through self-report questionnaires, which are instrumental for assessing subjective aspects of self-care behaviors. However, self-report data might be overestimated due to limitations such as memory errors and social desirability or recall biases. To address these issues, future studies could benefit from incorporating objective data sources that provide verifiable evidence of self-care behaviors. These could include physiological indicators such as blood pressure, blood sugar, and blood cholesterol levels, as well as medication adherence and diet records validated by monitoring devices like intelligent pill bottles [86]. Cross-referencing these objective measures with self-report questionnaires would help ensure the accuracy and reliability of the findings.

Point 5. Comorbidity and Health Status Data

While the manuscript includes comorbidity and health status data, the impact of these factors on the different domains of self-care behavior (medication, eating habits, lifestyle) is not fully explored.  Please include a more detailed analysis of how comorbidities, such as diabetes or hypertension, affect stroke self-care behaviors. This will provide a more nuanced understanding of the factors influencing self-care beyond self-efficacy and health literacy.

Response 5: Thank you for your thorough review and insightful comments regarding the analysis of comorbidity and health status data in our study. Based on your feedback, we have included a more detailed examination of how comorbidities such as hypertension, diabetes, and hyperlipidemia specifically affect self-care behaviors among stroke patients. We have expanded our analysis to highlight the significant role of hypertension as both a risk factor and a catalyst for better self-care behaviors due to increased patient awareness and knowledge. Additionally, we have discussed the challenges of medication adherence and the paramount importance of lifestyle modifications in managing stroke risk and recurrence.

Page 11-12, lines 352-377

This study found that hypertension significantly influenced lifestyle self-care behavior. Patients with hypertension showed better stroke behaviors than those without hypertension. According to Kao et al. [61], comorbidities, including hypertension, significantly affect self-care and functional recovery in stroke patients. Similarly, Wardhani [62] reported that hypertension, diabetes, and hyperlipidemia negatively impact self-care and NIHSS score improvements. Hypertension was the most common comorbidity among participants (40.2 %), consistent with previous studies [49,63,64]. It is a significant risk factor for stroke, affecting approximately 70% of stroke patients [65]. This condition increases stress on the vascular walls, causes endothelial dysfunction, stiffens large arteries in the brain, and contributes to atherosclerosis and carotid artery disease, thereby increasing the likelihood of stroke [66]. Approximately 20% of all ischemic stroke cases occur in patients with hypertension [65]. Living with hypertension also leads to greater awareness and knowledge of the condition, resulting in better self-care behaviors [67]. The study found that the average stroke duration among the subjects was 3.12±4.10 years. This indicates that individuals may be more proactive in recognizing and managing their highest risk during a relatively short timeframe. Additionally, the management of hypertension is considered to be more straightforward compared to diabetes or hyperlipidemia, possibly because patients can more easily address it through medication and lifestyle adjustments [68,69]. Additionally, lifestyle modification and management effectively prevent stroke compared to other methods that treat stroke risk factors [22,48]. However, some stroke survivors mistakenly believe that taking preventive medication is more essential than modifying lifestyle behaviors [70]. In spite of this, medication nonadherence is high among stroke patients [71]. Thus, lifestyle management plays a critical role in stroke prevention and recurrence [48]. Therefore, improving stroke patients' self-care abilities requires a comprehensive approach that emphasizes the need for lifestyle changes along with other self-care areas.

Point 6. Confounding Variables

The study does not adequately account for potential confounding variables such as socio-economic status, caregiver support, or healthcare access, which could affect both self-efficacy and self-care behaviors. Please include a discussion of potential confounding variables and how they may have influenced the results. A more robust multivariate analysis that accounts for these factors could strengthen your conclusions.

Response 6: Thank you for your excellent suggestions regarding the consideration of confounding variables in our study. Based on your insightful feedback, we have elaborated on how these factors could have influenced our results in the revised manuscript. We have discussed relevant studies that suggest various impacts of these variables on stroke patients' self-care behaviors and highlighted the need for incorporating these considerations into future research.

Page 12-13, lines 416-434

In this study, we conducted a comprehensive review of previous studies and incorporated variables deemed to influence the self-care behavior of stroke patients. We assessed the impact of these variables on self-care; however, it's critical to note that not all potential influencing factors were considered, nor were the interrelationships between self-care and other variables thoroughly explored. Wang et al. [81] discovered that not only did stroke patients' self-care improve, but caregivers' proactive attitudes also had a positive impact. Additionally, Peyman [82] affirmed that clear communication and easily understandable educational materials enhanced patients' self-care behavior. Moreover, Shuqi et al. [79] and Chau et al. [83] found that improved access to medical care was associated with greater self-efficacy and self-care behavior. Conversely, Babkair & Dickson [84] established that low socioeconomic status and a lack of family and social support were limiting factors for self-care behavior. In Korea, family members traditionally care for patients due to the country's medical system and culture [85]. Additionally, the National Health Insurance Service covers 97% of the population, ensuring high accessibility to medical care for all citizens [55]. However, the significant out-of-pocket expenses (approximately 65%) may impede medical accessibility for low-income people [55]. Future studies should consider analyzing variables related to caregivers' attitudes, stroke education, social status, medical care accessibility, and support from family and society to strengthen the conclusions of this study.

Point 7. Measurement Tools

Some of the tools used to measure variables, particularly health beliefs, have lower reliability scores (Cronbach’s α = 0.41 to 0.84), raising concerns about consistency. Please discuss the reliability limitations of the measurement tools and consider recommending alternative or additional validated instruments in future research.

Response 7: Thank you for your comprehensive review and your comments concerning the measurement tools used in our study. In response to your feedback, we have expanded our discussion on the limitations related to the reliability and validity of these tools within the manuscript. We have specifically addressed the issues with the tools that have not been validated for stroke patients and the low-reliability scores that may affect the consistency of the data collected. Moreover, we have provided explanations for the low reliability observed and suggested practical measures to address these limitations.

Page 13, lines 448-460

Thirdly, the mobile health literacy tool used in this study was validated for Korean adults but not stroke patients, and the other tools' validity still needs to be confirmed. Previous studies on domestic stroke patients used tools other than mobile health literacy. Although expert validity was assessed during the study, establishing the tool's validity may not be sufficient. Therefore, future research should utilize tools that have been validated for domestic stroke patients. Additionally, the sensitivity reliability was low at 0.41 when measuring health beliefs in this study. This could be attributed to the tool's limited number of items or the survey being conducted during outpatient treatment and hospitalization for first-episode stroke patients [87], reflecting the participants' emotional distress and potential misunderstanding of the survey items [88,89]. To address this, future studies could consider utilizing a mobile survey method so that patients can respond in a more stable environment after returning home.

Point 8. Detailed Intervention Suggestions

While the study provides useful findings, the manuscript lacks practical recommendations on how these insights can be translated into clinical or public health interventions. Please expand the discussion to include actionable recommendations for healthcare professionals and policymakers. For example, you could suggest specific mobile health interventions or educational programs that target health literacy and self-efficacy.

Response 8: Thank you for your careful review of this study's extensibility. Based on your comments, we have included practical recommendations for future research, as described below.

Page 14, lines 483-488

Continuous management is crucial for patients with conditions like stroke. Developing a customized mobile app could help ensure ongoing rehabilitation and self-care behaviors. This approach would be particularly suitable in Korea, given its well-established mobile service environment and high smartphone usage. Moreover, by providing the service through hospitals, we can anticipate improved health management effectiveness and reduced medical costs by using health data beyond hospital settings.

Point 9. Underreporting of Negative or Non-Significant Results

The manuscript focuses on positive relationships between variables, such as self-efficacy and self-care behaviors, but it does not adequately address non-significant findings. Please include a discussion of variables that did not show significant relationships and explore potential reasons for these findings. This will provide a more balanced interpretation of the results.

Response 9: Thank you for your suggestions for a balanced interpretation of this study. In the analysis results, we added descriptions of sex, the number of comorbidities, stroke-related health knowledge, and health beliefs that were entered into the regression analysis model but did not show statistical significance. We also described the reasons why no significant relationship was shown.

Page 14, lines 378-415

Contrary to previous research, the regression analysis in this study did not find a significant relationship between specific participant characteristics and self-care behavior. Sex was not identified as a substantial factor influencing self-care behavior, aligning with findings from Sun et al. [72], Delgado et al. [73], and Dawson et al. [74]. However, Jirjees et al. [75] revealed that women had a higher level of stroke knowledge than men, which could impact self-care behavior. Venditti et al. [76] and Arapi et al. [77] reported conflicting results, suggesting that men had higher self-care behaviors. These discrepancies may be due to differences in research methods, target groups, and cultural influences, highlighting the need for further investigation into gender differences. While previous research indicated that patients with comorbidities had lower self-care behavior levels [72], our study found that the number of comorbidities did not significantly affect self-care behavior. This result aligns with studies by Park & Kim [63] and Kuo et al. [78]. Notably, the presence of hypertension was found to influence stroke self-care behaviors, suggesting the need for a nuanced approach considering the type, severity, and frequency of diseases rather than just the number of comorbidities. Stroke-related health knowledge was also not identified as a significant factor influencing self-care behavior. While patients with increased health education and knowledge may exhibit better self-care behavior [79], the study did not compare the disease-related education experience before and after stroke diagnosis. This limited the confirmation of the impact of knowledge on self-care behavior. Additionally, although 70.8% of the study participants had general knowledge about stroke, only about 20% knew all the symptoms and risk factors [75]. The average stroke-related health knowledge score was 19.17 out of 25 points, but it's important to note that the overall knowledge of stroke was measured using only 25 questions. Furthermore, 36.4% of the participants had experienced a stroke in less than one year, indicating that the focus was mainly on acute treatment, and the overall level of knowledge about stroke was low, which did not significantly affect self-care behavior. The study found that health beliefs did not considerably impact self-care behavior, which aligns with a study by Ryu et al. [10], which focused on patients admitted to nursing hospitals. It was noted that the patient's health beliefs may not have influenced their self-care behavior, possibly due to the limited scope of their actions. This study included recently admitted patients, which could partially explain the results. Additionally, it was observed that behavioral patterns and health beliefs in patients with chronic diseases begin to take shape within a few months after diagnosis and progress steadily [80]. The average stroke duration in the two studies' participants was 2.34 years in Ryu et al. [10] and 3.12 years in this study. In addition, 36.4% (n = 36) of patients had a stroke within one year of onset, and 83.8% (n = 83) of patients had a stroke less than five years of onset, which means that many subjects were in the early stages of the disease. It can be concluded that this was a period when health beliefs about stroke were not sufficiently formed.

Point 10. Over-Estimation of Stroke Self-Efficacy Impact

The manuscript may overestimate the influence of stroke self-efficacy, without fully considering other contextual factors like family support or access to healthcare. Please include a more balanced interpretation by acknowledging that while self-efficacy is important, other external factors could significantly affect self-care behaviors. A more nuanced discussion of these factors will provide a deeper understanding of stroke management.

Response 10: Thank you for your input regarding a deeper understanding of stroke patient management in this study. The study explored factors affecting stroke self-care, but it did not cover all relevant factors. As per your suggestion, we have included additional information on factors that may impact self-care behavior and on confounding variables that may affect self-efficacy and self-care behavior.

Page 12-13, lines 416-434

In this study, we conducted a comprehensive review of previous studies and incorporated variables deemed to influence the self-care behavior of stroke patients. We assessed the impact of these variables on self-care; however, it's critical to note that not all potential influencing factors were considered, nor were the interrelationships between self-care and other variables thoroughly explored. Wang et al. [81] discovered that not only did stroke patients' self-care improve, but caregivers' proactive attitudes also had a positive impact. Additionally, Peyman [82] affirmed that clear communication and easily understandable educational materials enhanced patients' self-care behavior. Moreover, Shuqi et al. [79] and Chau et al. [83] found that improved access to medical care was associated with greater self-efficacy and self-care behavior. Conversely, Babkair & Dickson [84] established that low socioeconomic status and a lack of family and social support were limiting factors for self-care behavior. In Korea, family members traditionally care for patients due to the country's medical system and culture [85]. Additionally, the National Health Insurance Service covers 97% of the population, ensuring high accessibility to medical care for all citizens [55]. However, the significant out-of-pocket expenses (approximately 65%) may impede medical accessibility for low-income people [55]. Future studies should consider analyzing variables related to caregivers' attitudes, stroke education, social status, medical care accessibility, and support from family and society to strengthen the conclusions of this study.

Point 11. Literacy Over-Estimation

The impact of mobile health literacy on medication adherence may be overestimated, given that general health literacy and access to digital resources are not fully addressed. Consider expanding the discussion to explore how general health literacy, digital accessibility, and socioeconomic factors may moderate the relationship between mobile health literacy and self-care behavior.

Response 11: We appreciate your opinion on the potential overestimation of mobile health literacy's effect, especially considering the influences of general health literacy, digital accessibility, and socioeconomic factors. In response to your feedback, we have expanded the discussion within our manuscript to more comprehensively address these moderating factors. We have included a detailed analysis of the general health literacy and digital infrastructure in Korea, as well as the implications of socioeconomic conditions on the utilization of digital health resources. By doing so, we aimed to provide a more balanced view of the factors influencing mobile health literacy and its relationship with self-care behaviors.

Page 12, lines 334-351

This study did not adequately account for health literacy, digital resource accessibility, and socioeconomic factors, which suggests that the impact of mobile health literacy on medication self-care behaviors may have been overestimated. Therefore, it is essential to exercise caution when interpreting the results. However, it's worth noting that in Korea, health literacy and digital resource accessibility are excellent. The country ensures medical accessibility through National Health Insurance and medical protection programs, allowing people to access health information easily [55]. With an illiteracy rate of less than 1% [56], the overall health literacy in Korea is high. Moreover, Korea boasts one of the world's best Internet infrastructures [57], and 93.0% of the population uses smartphones [58], indicating a robust digital infrastructure. Nevertheless, previous research has shown that individuals with high health literacy tend to exhibit better self-care behaviors [59]. In contrast, those with low mobile literacy may struggle to utilize mobile-based services effectively [21]. Sieck et al. [21] highlighted how limitations in digital accessibility can have a negative impact on self-care behavior, and Smith & Magnani [60] emphasized the significant influence of socioeconomic factors on patients' ability to use digital devices and access medical services. Therefore, it's essential to approach the findings of this study with caution, and future research should consider these factors in its design.

Reviewer 2 Report

Comments and Suggestions for Authors

After a stroke, recovery and the return to a full life are the main goals for survivors, their families, and healthcare professionals who strive to provide the best possible care. This is achieved, among other measures, through neurorehabilitation. Thus, the need for effective rehabilitation after a stroke becomes an essential element in the continuity of care and attention that should be provided to these patients. The objective of this work is to examine the influence of mobile health literacy, stroke-related health knowledge, health beliefs, and self-efficacy on self-care behaviors among stroke patients.

Here there are some suggestions:

1/ In the Introduction, the proposed objectives are defined in an excessively generic format (92-93), which does not sufficiently differentiate them from the evidence presented in the Introduction (83-90). These are very broad concepts, difficult to reproduce and measure in the case of replicating similar studies:  “This study examined the influence of mobile health literacy, stroke-related health 91 knowledge, health beliefs, and self-efficacy on…… “

2/ The method used in the selection of patients is not described.

3/ In measurements: The severity of the stroke is not included in the inclusion criteria, which, combined with the small sample size, may affect the results. I consider it essential to include a description of the stroke severity in the patients included in the study and the value of its magnitude in relation to the total number of strokes. This will allow for the evaluation of the impact of potential interventions on this stroke subgroup in relation to the total stroke incidence.

4/ Have the scales used in the assessment of stroke self-care (Kang's tool), mobile health literacy (Norman and Skinner), stroke-related health knowledge (Rehe et al.), health beliefs (Becker's Health Belief Model), and stroke self-efficacy (Kang and Yoon) been previously validated and adapted to the specific pathology of stroke? Their use constitutes a limitation in the validation of the results since some are only associated with previous descriptive studies.

Author Response

Response to Reviewer 2 Comments

We appreciate the time and effort the reviewer has put into the valuable feedback and insightful comments on this manuscript. We have carefully considered each comment and made changes to the manuscript, as required. We have marked the revisions made to the manuscript in blue font.

Point 1. In the Introduction, the proposed objectives are defined in an excessively generic format (92-93), which does not sufficiently differentiate them from the evidence presented in the Introduction (83-90). These are very broad concepts, difficult to reproduce and measure in the case of replicating similar studies: “This study examined the influence of mobile health literacy, stroke-related health 91 knowledge, health beliefs, and self-efficacy on…… “

Response 1: Thank you for your constructive feedback regarding the clarity and specificity of the objectives outlined in our introduction. In response to your comments, we have revised the objectives in the manuscript to specify the distinct roles that mobile health literacy, stroke-related health knowledge, health beliefs, and self-efficacy play in influencing self-care behaviors among stroke patients.

Page 2, lines 91-99

This study has practical implications for stroke patient care and intervention programs. This study aims to delineate the specific impacts of mobile health literacy, stroke-related health knowledge, health beliefs, and self-efficacy on the self-care behaviors of stroke patients. By rigorously analyzing how each of these factors independently and interactively influences self-care, we seek to provide precise, actionable insights that can directly inform the development of targeted self-care intervention programs. Additionally, leveraging our findings, we suggest designing and piloting a mobile-based self-care intervention program tailored specifically to enhance the self-management capabilities of stroke survivors.

Point 2. The method used in the selection of patients is not described.

Response 2: Thank you for your thorough review and for pointing out the omission of the patient selection criteria in our initial manuscript. In response to your feedback, we have thoroughly described the selection process for our study participants in the revised section of our manuscript. We have included both inclusion and exclusion criteria to ensure a clear understanding of how participants were chosen and the specific characteristics that qualified them for our study.

Page 3, lines 108-124

2.2 Study Population and Sampling

This study involved stroke patients diagnosed and treated at three general hospitals in City C. All outpatients and inpatients were invited to participate in rehabilitation treatment and follow-up observation during the recruitment period. The specific selection criteria were as follows: 1) patients diagnosed with ischemic stroke who had been taking antithrombotic medication for at least one month, 2) patients who were able to communicate and complete a questionnaire, 3) patients who understood the purpose of the study and willingly agreed to participate, and 4) adults aged 19 years or older, 5) patients with first-episode stroke with a National Institutes of Health Stroke Scale (NIHSS) score of 10 or more. The NIHSS score was determined based on Xing & Wei [31], who indicated that patients with a score of 10 or more had the cognitive and physical capacity to complete the self-report questionnaire. Additionally, based on previous research showing differences in self-care behaviors and focus between first-episode and recurrent stroke patients [32], this study focused solely on first-episode patients. Patients diagnosed with hemorrhagic stroke, those hospitalized due to complications from other underlying diseases, those with cognitive impairments who could not understand the questionnaire and perform self-care, and those who did not fully respond to the survey items were excluded from the study.

Point 3. In measurements: The severity of the stroke is not included in the inclusion criteria, which, combined with the small sample size, may affect the results. I consider it essential to include a description of the stroke severity in the patients included in the study and the value of its magnitude in relation to the total number of strokes. This will allow for the evaluation of the impact of potential interventions on this stroke subgroup in relation to the total stroke incidence.

Response 3: Thank you for your insightful feedback concerning the inclusion of stroke severity in our study criteria. We acknowledge the omission of the NIHSS score as a criterion in our initial manuscript and appreciate your emphasis on its importance for accurately characterizing the stroke severity among participants. In response to your comments, we have revised our manuscript to include the NIHSS score as a criterion for patient recruitment. We have also detailed how this scale, which assesses stroke severity, allows us to effectively represent and study a substantial segment of the stroke patient population.

Page 10, lines 291-302

The study participants were chosen based on an NIHSS score of 10 or more, as indicated by Xing & Wei [31], who found that patients within this range possessed the cognitive and physical capacity necessary to complete the self-report questionnaire. This study specifically focused on patients experiencing their first stroke episode, taking into account the differing self-care behaviors and focusing on first-episode and recurrent stroke patients [32]. Saber & Saver [45] reported that approximately 71% of stroke hospitalizations in the United States are attributed to patients with an NIHSS score of 0-4, while only 29% are attributed to those with a score of 10 or higher. Despite the small sample size and convenience sampling, the participants of this study can be considered representative of stroke patients. Therefore, the factors influencing self-care behavior identified in this study can be deemed sufficiently viable as foundational data for developing interventions for stroke patients.

Point 4. Have the scales used in the assessment of stroke self-care (Kang's tool), mobile health literacy (Norman and Skinner), stroke-related health knowledge (Rehe et al.), health beliefs (Becker's Health Belief Model), and stroke self-efficacy (Kang and Yoon) been previously validated and adapted to the specific pathology of stroke? Their use constitutes a limitation in the validation of the results since some are only associated with previous descriptive studies.

Response 4: Thank you for your thorough review and insightful observations regarding the measurement tools used in our study. Given the absence of validated tools for Korean stroke patients, we opted for instruments that had been validated by experts in related fields. We recognize this as a limitation and have addressed it within the study by discussing the potential impact on the validation of our results. We are committed to enhancing the robustness of our findings and suggest that future research should focus on developing and validating tools tailored to the specific needs of stroke patients in Korea.

Page 13, lines 448-454

Thirdly, the mobile health literacy tool used in this study was validated for Korean adults but not stroke patients, and the other tools' validity still needs to be confirmed. Previous studies on domestic stroke patients used tools other than mobile health literacy. Although expert validity was assessed during the study, establishing the tool's validity may not be sufficient. Therefore, future research should utilize tools that have been validated for domestic stroke patients.

Round 2

Reviewer 1 Report

Comments and Suggestions for Authors

Thank you for addressing comments

Reviewer 2 Report

Comments and Suggestions for Authors

The modifications made by the authors are in line with the suggestions made by the reviewer, so I consider the manuscript Acceptable.